# Angiogenesis–Browning Interplay Mediated by Asprosin-Knockout Contributes to Weight Loss in Mice with Obesity

**DOI:** 10.3390/ijms232416166

**Published:** 2022-12-18

**Authors:** Tingting Yin, Sheng Chen, Guohua Zeng, Wanwan Yuan, Yanli Lu, Yanan Zhang, Qianqian Huang, Xiaowei Xiong, Baohua Xu, Qiren Huang

**Affiliations:** 1Key Provincial Laboratory of Basic Pharmacology, Nanchang University, Nanchang 330006, China; 2Department of Pharmacology, School of Pharmacy, Nanchang University, Nanchang 330006, China; 3Jiangxi Province Key Laboratory of Laboratory Animal, Nanchang 330006, China

**Keywords:** asprosin, obesity, adipocyte, angiogenesis, browning

## Abstract

Asprosin (ASP) is a recently identified adipokine secreted by white adipose tissue (WAT). It plays important roles in the maintenance of glucose homeostasis in the fasting state and in the occurrence and development of obesity. However, there is no report on whether and how ASP would inhibit angiogenesis and fat browning in the mouse adipose microenvironment. Therefore, the study sought to investigate the effects of *ASP*-knockout on angiogenesis and fat browning, and to identify the interaction between them in the ASP-knockout mouse adipose microenvironment. In the experiments in vivo, the *ASP*-knockout alleviated the obesity induced by a high fat diet (HFD) and increased the expressions of the browning-related proteins including uncoupling protein 1 (UCP1), PRD1-BF-1-RIZ1 homologus domain-containing protein-16 (PRDM16) and PPAR gamma coactivator 1 (PGC1-α) and the endothelial cell marker (CD31). In the experiments in vitro, treatment with the conditional medium (CM) from *ASP*-knockout adipocytes (*ASP^−/−^*-CM) significantly promoted the proliferation, migration and angiogenesis of vascular endothelial cells, and increased the expressions of vascular endothelial growth factor (VEGF)/vascular endothelial growth factor receptor 2 (VEGFR2) and phosphatidylinositol 3-kinase (PI3K)/protein kinase B (AKT)/endothelial nitric oxide synthase (eNOS) pathway proteins. In addition, the treatment with CM from endothelial cells (EC-CM) markedly reduced the accumulation of lipid droplets and increased the expressions of the browning-related proteins and the mitochondrial contents. Moreover, the treatment with EC-CM significantly improved the energy metabolism in 3T3-L1 adipocytes. These results highlight that *ASP*-knockout can promote the browning and angiogenesis of WAT, and the fat browning and angiogenesis can interact in the mouse adipose microenvironment, which contributes to weight loss in the mice with obesity.

## 1. Introduction

Obesity is a chronic inflammation characterized as an increase in number (hyperplasia) and/or an enlargement in size of adipocytes (hypertrophy). Data from evidence-based medicine have shown that obesity caused by the imbalance of energy homeostasis is commonly considered as an independent risk factor of cardiovascular diseases (CVD), including hypertension, stroke and atherosclerosis [1,2,3]. Accordingly, it would be helpful for the prevention and therapy of obesity-related CVD to elucidate the crosstalk between adipocytes and vascular cells in the adipose tissue microenvironment. It is generally thought that adipose tissue mainly includes white adipose tissue (WAT) and brown adipose tissue (BAT) in mammals [4]. WAT stores energy in the form of triglyceride (TG), while BAT dissipates energy in the form of heat. To a certain extent, BAT has natural anti-obesity effects [5]. Numerous studies have shown a negative correlation between BAT activation and body mass index (BMI), adipose tissue mass, and insulin resistance [6]. Therefore, the browning of WAT and/or BAT activation would provide a new strategy for the treatment of obesity.

The adipose tissue microenvironment is composed of cells and extracellular components. The former includes all kinds of cells, such as adipocytes, vascular cells, neurons, immune cells, and so on, while the latter covers hormones, growth factors, cytokines and the extracellular matrix secreted by these cells. In the microenvironment, the components described above show a balanced interplay and cooperatively regulate the homeostasis of the adipose tissue microenvironment [7,8]. For instance, the crosstalk between the adipocytes and endothelial cells (EC) plays critical roles in the maintenance of the homeostasis [9]. It is believed that the vessel density in WAT is much less than that in BAT, demonstrating that certain putative cytokines (adipokines) secreted by white adipocytes could inhibit angiogenesis [10]. Previous studies revealed that leptin induces angiogenesis, vascular permeability, and vascular remodeling [11]. In sharp contrast, adiponectin represses the angiogenesis [12].

Among the adipokines, ASP is a newly identified peptide adipokine secreted by WAT [13]. It plays important physiological roles in the control of glucose homeostasis, especially in the fasting state. ASP is also a centrally acting appetite-stimulating hormone. It increases appetite and ultimately leads to obesity and weight gain [14]. In addition, many studies have demonstrated that ASP is involved in the pathophysiological process of numerous metabolism disorders, including obesity, insulin resistance, type 2 diabetes mellitus, and so forth [15]. Wang et al. found that circulating ASP levels were significantly higher in patients with diabetes and obesity than in normal controls [16]. Romere et al. and Greenhill et al. reported that insulin-resistant patients and mice presented pathologically elevated ASP levels. Moreover, the inhibition of ASP expression by immunological or genetic means has shown better effects in lowering circulating glucose and insulin [13,15]. Many clinical studies have also found that the circulating ASP levels are positively correlated with the degree of obesity. Studies by Wang et al. have shown that serum ASP levels are higher in children with obesity and insulin resistance than in healthy children of normal weight [17]. However, to date, there is fewer research on the role of ASP in the pathogenesis of cardiovascular system. In particular, there is no report on whether and how ASP would impact angiogenesis and adipose tissue-browning in the adipose microenvironment.

Although there have been several studies on the relation between the angiogenesis and the adipose-browning [18], so far it remains unclear whether the crosstalk between the *ASP*-mediated angiogenesis and browning in the adipose microenvironment could promote each other to alleviate the metabolic disorders induced by HFD. Therefore, this study sought to explore the effects and interactions of ASP on the angiogenesis and adipose-browning in the adipose microenvironment in *ASP*-knockout cells and mouse models. The findings would provide new molecular targets and diagnostic molecular markers for the diagnosis and treatment of clinical obesity and complications. More importantly, a better understanding of the crosstalk between vasculature and adipose tissue would provide a novel approach to the treatment of obesity and obesity-related CVD.

## 2. Results

### 2.1. ASP-Knockout Alleviates Mouse Obesity Induced by HFD

To investigate the effect of ASP on obesity, we constructed *ASP* conditional knockout mice (CKO, Appendix A). Both CKO and WT mice were fed with HFD for 16 weeks. Although numerous studies in the literature have reported that ASP enhances appetite, our results exhibited that *ASP*-CKO failed to affect the food intake in mice significantly, indicating that the reductions in body weights caused by *ASP*-knockout do not result from the reduction of appetite (Appendix A). The results showed that, compared with the mice in the WT-ND group, the mice in the WT-HFD group manifested an obesity phenotype, including obesity appearances (Figure 1A), increased body weights (Figure 1B), Lee’s index (similar as human BMI) (Figure 1D) and visceral fat weights (Figure 1E,F), enlarged adipocyte sizes (Figure 1E,G) and unchanged body lengths (Figure 1C). However, there were significant decreases in certain parameters, including the body weights (Figure 1B), eWAT weights (Figure 1E,F) and adipocyte sizes (Figure 1E,G), except for the body length (Figure 1C) between the WT-ND and CKO-ND groups. Importantly, compared with the WT-HFD group, all the above parameters, except for body length, in the CKO-HFD group were significantly decreased, indicating that ASP-knockout alleviates the mouse obesity induced by HFD (Figure 1A–G).

### 2.2. ASP-Knockout Promotes the Angiogenesis and Browning of White Adipose In Vivo

It was reported that the angiogenesis in WAT would contribute to fat browning, which is beneficial for weight loss; therefore, we next evaluated the effect of *ASP*-knockout on the browning and angiogenesis in mouse WAT. The results of immunohistochemistry and immunofluorescence revealed that *ASP*-knockout, either with ND or HFD feeding, markedly increased the in situ expressions of both UCP1 (a thermogenic marker) and CD31 (an endothelial cell marker) in WAT compared with the WT mice (Figure 2A–C). The Western blot assay also showed similar results in the CD31 expression of mouse WAT (Figure 2D,E), indicating that *ASP*-knockout promotes angiogenesis in WAT. Next, we examined the fat browning by detecting the browning-related protein levels, including PRDM16, PGC1-α and UCP1.The Western blot results displayed that *ASP*-knockout significantly elevated the levels of these browning-related proteins, regardless of ND or HFD-feeding (Figure 2F,G), suggesting that *ASP*-knockout promotes the browning of WAT in mice. Moreover, the increased levels of UCP1 and CD31 exhibited a parallel change, implying that the browning and angiogenesis induced by *ASP*-knockout produces an interactive promotion in WAT.

### 2.3. ASP-Knockout Promotes the Migration and Angiogenesis of Vascular Endothelial Cells In Vitro

As described above, since the angiogenesis and fat browning mediated by *ASP*-knockout could promote each other, we speculated that the interaction between the endothelial cells and the adipocytes could play crucial roles in the maintenance of energy metabolism homeostasis in the adipose microenvironment. To verify this, we first observed the effect of the CM from adipocytes on the migration and angiogenesis in HUVECs. We successfully established *ASP*-knockout 3T3-L1 preadipocytes (*ASP^−/−^*) by CRSPR-Cas 9 technology (Appendix A; Figure 3A,B). Our results displayed that *ASP*-knockout in 3T3-L1 preadipocytes did not influence the differentiation of preadipocytes compared with the WT preadipocytes (Appendix A). The data of the wound-healing assay showed that, compared with the WT-CM group, the treatment of HUVECs with *ASP^−/−^*-CM significantly elevated the 24 h and 48 h cell healing rates, respectively, demonstrating that *ASP*-knockout promotes the migration of endothelial cells (Figure 3C,D). Moreover, the results of the Matrigel tube formation assay displayed that treatment of HUVECs with *ASP^−/−^*-CM significantly elevated the numbers of junctions (Nb) and total lengths (Tot) of branches compared with the WT-CM group, demonstrating that *ASP*-knockout promotes angiogenesis in vitro (Figure 3E–G). *ASP*: asprosin.

### 2.4. ASP-Knockout Enhances VEGF/VEGFR2 and PI3K/AKT/eNOS Signaling In Vitro

Since VEGF/VEGFR2 signaling is involved in angiogenesis, we next evaluated the effects of *ASP*-knockout on the VEGF/VEGFR2 signaling. The Western blot results exhibited that the treatment of HUVECs with *ASP^−/−^*-CM significantly increased the expression levels of VEGF and VEGFR2 compared with the WT-CM group, demonstrating that *ASP*-knockout enhances the VEGF/VEGFR2 signaling (Figure 4A,B). Moreover, it was reported that the PI3K/AKT/eNOS pathway is not only involved in angiogenesis, but also regulates the function of endothelial cells. Consequently, we tested the effects of *ASP*-knockout on the PI3K/AKT/eNOS pathway. As anticipated, the results showed that the treatment of HUVECs with *ASP^−/−^*-CM significantly elevated the ratios of p-PI3K/PI3K, p-AKT/AKT and p-eNOS/eNOS, respectively, compared with the WT-CM group (Figure 4C,D). Moreover, we obtained similar expression results for the PI3K/AKT signaling in the aortic tissue of *ASP*-CKO mice (Appendix A). Taken together, these data indicate that *ASP*-knockout activates the PI3K/AKT/eNOS pathway.

### 2.5. Effect of EC-CM on Lipid Aggregation and Browning In Vitro

The results obtained so far have shown that adipocytes may regulate the biological function of the endothelial cell, including migration, angiogenesis and endocrine function by secreting adipokines such as ASP. Thus, an intriguing question is whether the endothelial cells would influence the biological function of the adipocytes, such as browning, by secreting some vaso-active substances. To address this question, we utilized the EC-CM to treat the 3T3-L1 adipocytes in order to observe the browning in vitro. The oil red O staining results showed that treatment of 3T3-L1 adipocytes with EC-CM markedly reduced the lipid accumulation compared with the normal medium (NM), suggesting that EC-CM treatment increases the consumption of lipid by means of a pro-browning effect (Figure 5A,B). To further verify whether the decreased lipid accumulation induced by the EC-CM treatment would be caused by pro-browning, we detected the expression levels of the browning-related proteins. As expected, the results revealed that the EC-CM treatment considerably elevated the expression levels of the browning-related proteins, including UCP1, PGC1-α and PRDM16, demonstrating that the EC-CM treatment promotes the browning of white adipocytes (Figure 5C,D).

### 2.6. EC-CM Treatment Enhances the Mitochondrial Biogenesis and Respiratory Function of Adipocytes In Vitro

Since brown adipocytes have a larger number of mitochondria than white adipocytes, we examined the effects of EC-CM treatment on the mitochondrial contents using a Mito-Tracker assay. The results showed that the treatment of 3T3-L1 adipocytes with EC-CM for 48 h markedly enhanced the fluorescence intensity compared with the NM treatment, demonstrating that the EC-CM treatment enhances the mitochondrial contents (Figure 6A,B). Finally, we investigated the effects of the EC-CM treatment on mitochondrial respiration using a Seahorse XF analyzer. The present results revealed that EC-CM treatment for 48 h enhanced not only the basal, but also the stressed respiration function of mitochondria, presented as increases in proton leakage, ATP production and OCR compared with the NM treatment. This indicates that the EC-CM treatment enhances the mitochondrial function (Figure 6C,D). Taken together, the present data demonstrate that the EC-CM treatment enhances the mitochondrial biogenesis and respiratory function, suggesting that the EC-CM treatment promotes the browning of 3T3-L1 adipocytes.

## 3. Discussion

Obesity is prevalent worldwide and is closely associated with the incidence of many cardiocerebrovascular diseases, such as stroke, hypertension and atherosclerosis. It is estimated that adipose tissue accounts for 20% of the body weight in healthy adults while it exceeds approximately 40% in individuals with obesity [19]. Adipose tissue, as a metabolically active organs, can undergo structural and cellular remodeling when exposed to different environments, such as diet and temperature challenges. Currently, the most common anti-obesity medication is supposed to limit energy intake by reducing appetite and/or reducing intestinal absorption. However, the medication often causes many adverse effects and seriously affects the life quality of the patients [20]. In recent years, a growing number of data have suggested that the activation of BAT-mediated adaptive non-shivering thermogenesis would be a plausible strategy to combat weight gain and maintain glucose homeostasis [21]. Considering the existence of the excess WAT in individuals with obesity, the transformation of WAT into BAT with thermogenic activity would be a promising therapeutic approach [22].

Adipose tissue is a highly vascularized tissue, in which the adipocytes are nourished by an extensive capillary network [23]. Moreover, as an endocrine gland, adipose tissue secretes a large amount of hormones, growth factors and cytokines, which regulates angiogenesis. When suffering from the challenge of energy status, adipose tissue would undergo constant remodeling [24]. WAT with a low metabolic activity exhibits lower vessel densities than BAT with a high metabolic activity [25]. Accordingly, angiogenesis in BAT is supposed to be more efficient compared to that in WAT. Additionally, browning of WAT is often accompanied by angiogenesis during cold exposure [26]. Increasing evidence indicates that the vascularization of adipose tissue would become a new target for the treatment of obesity and related metabolic disorders [27,28]. In this study, we found, for the first time, that *ASP*-knockout can promote white fat browning and angiogenesis, and they can promote each other both in vitro and in vivo.

It was reported that, in people with obesity, the vessel density in the adipose microenvironment is decreased, but the circulating ASP level is increased [29,30]. To explore whether there would be a causal relationship between the reduced ASP level and angiogenesis in the adipose microenvironment, we successfully established *ASP*-conditional knockout mice (CKO mice) by gene-editing technology (Appendix A). Our present results showed that the CKO mice had an extremely low circulating level of ASP and manifested an anti-obesity phenotype. In particular, the CKO mice exhibited more potent angiogenesis in eWAT than the WT mice, characterized by the elevated expression of CD31, a marker of endothelial cells [31], suggesting that *ASP*-knockout is beneficial for the angiogenesis in WAT in vivo.

It has been demonstrated that there is a strong positive link between the vessel density and fat browning, and pro-angiogenesis can induce fat browning in WAT [32]. An important hallmark of fat browning is the enhancement of adaptive thermogenesis mediated by UCP1. UCP1 is localized at the inner membrane of mitochondria and is highly expressed in brown and beige adipocytes [33]. In the present study, we examined the expressions of UCP1 and other browning-related proteins, such as PRDM16 and PGC1-α, in WAT of the *ASP*-knockout mice by immunohistochemistry and Western blots. Our data revealed that the increased expressions of UCP1, PRDM16 and PGC1-α were accompanied by the increased expressions of CD31 in the CKO mouse WAT, indicating that there would be a positive relationship between the browning-related proteins and CD31. The evidence further verified that the *ASP*-knockout promotes angiogenesis and fat browning in WAT.

Next, the question of whether there a mutual causal relation would exist between the pro-angiogenesis and the pro-browning mediated by *ASP*-knockout was raised. A previous study reported that the deletion of VEGF-A resulted in the ‘whitening’ of BAT, whereas the overexpression of VEGF-A brought about the ‘beigeing’ of WAT in mice [34]. Moreover, it was thought that the vascularization of WAT is a critical step in the browning of WAT [35]. To address the question, we successfully established the *ASP*^−/−^-3T3-L1 preadipocyte line by CRSPR-Cas 9 technology (Appendix A). After successful induction of differentiation, we collected the CM and treated the HUVECs with CM to observe the migration and angiogenesis in vitro. We found, for the first time, that ASP-knockout promoted the proliferation, migration and angiogenesis of the endothelial cells in vitro. VEGF/VEGFR2 signaling is a key pathway to regulate angiogenesis [36]. VEGF binds to VEGFR2 on the membrane of endothelial cells and activates the intracellular MAPK/EKR pathway to promote cell proliferation, migration and angiogenesis. Moreover, the PI3K/AKT/eNOS/NO pathway plays critical roles in determining the endocrine function of endothelial cells [37]. AKT can promote eNOS phosphorylation and NO release, which activates its downstream pathway and increases VEGF expression in endothelial cells [38,39]. PI3K/AKT/eNOS and VEGF/VEGFR2 are the key pathways in endothelial proliferation, migration and angiogenesis. Our data elucidated that the *ASP*-knockout treatment increased the expressions of VEGF and VEGFR2 and activated the PI3K/AKT/eNOS pathway. These results suggest that *ASP*-knockout contributes to angiogenesis in the adipose microenvironment.

Considering that *ASP*-knockout promoted angiogenesis in the adipose microenvironment both in vivo and in vitro, another question of whether the angiogenesis in the adipose microenvironment could promote the browning of adipocytes arose. Although we have elucidated that angiogenesis and browning were enhanced, and the enhancement exhibited synchronicity in the WAT of *ASP*-knockout mice, we directly explored it using the EC-CM. The ECs are considered as endocrine cells which secrete large numbers of vaso-active substances, such as NO and endothelin 1, as well as mitogens, including FGFs, VEGF, and so on. Some studies have shown that VEGF secreted by endothelial cells could promote the fat browning and, therefore, affect the lipid levels [40,41]. As a result, in the study, we used the EC-CM as a pro-browning stimulant. We found that the EC-CM reduced the lipid accumulation, increased the expressions of the browning-related proteins (UCP1, PGC1-α and PRDM16) and the mitochondrial contents, with energy metabolism of mitochondria enhanced. As a result, we concluded that EC-CM promoted the browning of adipocytes, which was consistent with the results reported by Shimizu and Huber [42,43].

Of course, there were some flaws in the study. Although we explored the crosstalk between the angiogenesis and the fat browning mediated by *ASP*-knockout both in vivo and in vitro, the results in vitro were not particularly a physical crosstalk between the ECs and adipocytes, but rather an indirect crosstalk, due to use of the conditional medium. In fact, a better approach for it should be to use a 3-D cell culture system. In addition, although we evaluated the mitochondrial contents by a relatively sensitive Mito-Tracker using confocal microscopy, this method only presents the relative contents of mitochondria. A better choice could be transmission electronic microscopy. Finally, as for the study on energy metabolism, we only detected OCR in vitro, but OCR in vitro did not authentically represent the in vivo state. A better alternative to assess the energy metabolism level could be the metabolism cage test in vivo. All these limitations need to be addressed in future studies.

In summary, our present study investigated, for the first time, whether and how *ASP*-knockout would affect the angiogenesis and fat browning, as well as their crosstalk, in the adipose microenvironment, both in vivo and in vitro. Our present results elucidated that *ASP*-knockout can promote the angiogenesis and browning of WAT, and the angiogenesis and browning can interactively be promoted in the adipose microenvironment. These findings could provide strong evidence that ASP is involved in the process of obesity and could offer novel targets for the treatment of obesity and related cardiovascular diseases.

## 4. Materials and Methods

### 4.1. Animal Experimental Protocols

All animal experiments were approved by the Institutional Animal Care and Use Committee of Nanchang University School of Medicine and conducted in accordance with the Guide for the Care and Use of Laboratory Animals published by the US National Institute of Health (NIH Publication No.85-23, revised 1996). Twenty male wild-type (WT) C57BL/6 mice (SPF grade, Nanchang University Laboratory Animal Science Center) were randomly divided to WT-ND and WT-HFD groups in which the WT mice were, respectively, fed with a normal diet (ND) and a high fat diet (HFD, 60%, MedScience, Catelog#MD12033) for 16 weeks; meanwhile, 20 male adipose tissue-conditional knockout (CKO) C57BL/6 mice (SPF grade, Shanghai Model Orgaisms Center, Inc., Shanghai, China) were also randomly allocated to CKO-ND and CKO-HFD groups, in which the CKO mice were, respectively, fed with ND and HFD for 16 weeks, and there were 10 mice in each group. The body weights and lengths were measured, and the Lee’s index was calculated (Lee’s index = [Weight (g) × 10^3^/Body length (cm)] ^1/3^) [44].

### 4.2. Differentiation and Treatment of 3T3-L1 Preadipocytes

The differentiation of 3T3-L1 preadipocytes was performed following our previously published paper [45]. Briefly, WT-3T3-L1 preadipocytes (ATCC, Gaithersburg, MD, USA) and *ASP-*knockout 3T3-L1 preadipocytes (*ASP^−/−^*, GeneChem, Shanghai, China) were routinely cultured in H-DMEM (Solarbio, Beijing, China) supplemented with 10% FBS. Upon growing to 95% confluence, the 3T3-L1 preadipocytes were subjected to the induction of differentiation. The cells were first induced for the first 2 days with H-DMEM containing 3-isobutyl-1-methylxanthine (IBMX, 0.5 mM), insulin (INS, 10 µg/mL), dexamethasone (DEX, 1 µM), rosiglitazone (RGZ, 1 µM) and 10% FBS, and then differentiation was performed for the middle 6 days with H-DMEM containing INS (10 µg/mL) and 10% FBS, during which the differentiation medium was changed every two days. Finally, the cells were treated with the conditional medium of endothelial cells (CM-EC) containing 10% FBS for the last 2 days. The putative mature adipocytes were identified by Western blot analysis and oil red O staining.

### 4.3. Histological Analysis

Adipose tissue isolated from C57BL/6 mice was first fixed with 4% paraformaldehyde and embedded in paraffin after a series of gradient dehydration. The paraffin blocks were sliced into approximately 8 μm slices. For the H&E assay, de-paraffinized slices were stained with hematoxylin and eosin (H&E) for 3 min. For the immunohistochemistry assay, the de-paraffinized slices were first incubated with 3% H_2_O_2_ for 10 min to quench the endogenous peroxidase, and then incubated with the boiled sodium citrate buffer for 15 min to repair the antigens. Next, the slices were incubated with 5% BSA (BSA) for 10 min, followed by overnight incubation with UCP1-antibody at 4 °C. Subsequently, the slides were incubated with horseradish peroxidase (HRP)-conjugated secondary antibody at room temperature for 2 h. Finally, the slides were developed with a DAB developer for 15 min. For the immunofluorescence assay, except for the incubation with a fluorescent-conjugated anti-goat secondary antibody at 1:200 (Dylight 649, Proteintech, Wuhan, China), the other procedures were the same as the immunohistochemistry assay. The images were photographed under an inverted microscope (Olympus, Tokyo, Japan).

### 4.4. Wound-Healing Assay

HUVECs (ATCC, Gaithersburg, MD, USA) were seeded on a 6-well plate (2.5 × 10^5^ cells/well). The growing cells, growing at a logarithmic growth rate, were scratched with a 200 μL pipette tip. The scratched cells were, respectively, incubated in the conditional medium from WT-3T3-L1 adipocytes (WT-CM) and *ASP*^−/−^-3T3-L1 adipocytes (*ASP*^−/−^-CM) for 24 h or 48 h. The migration of HUVECs was detected by the wound-healing assay at 0 h, 24 h and 48 h from the scratch.

### 4.5. Matrigel Tube Formation Assay for Angiogenesis

The standard Matrigel (Corning, New York, NY, USA) assay was used to assess the spontaneous formation of capillary-like structures in vitro. Pre-cold 96-well plates were taken out from a refrigerator and 60 μL Matrigel gel was slowly added to the wells of the plates. HUVECs were inoculated on the 96-well plate pre-covered by the gel at a cell density of approximately 2.5 × 10^4^ cells/well. The cells were, respectively, incubated in WT-CM and *ASP*^−/−^-CM for 8 h. A tube-like structure was observed under an inverted microscope (Olympus, Tokyo, Japan). The number (Nb) of junctions and total (Tot) branching length of the microvascular network were calculated and photographed.

### 4.6. Oil Red O Staining and Quantification

The 3T3-L1 adipocytes treated with EC-CM were stained with oil red O as described in our previous reports [45]. Briefly, the cells were washed with PBS and fixed with 4.0% formaldehyde solution for 30 min, followed by staining with oil red O (Solarbio, Beijing, China) for 1 h. After staining, the cells were observed and photographed using an inverted microscope. Afterwards, the cells were treated with 100% isopropanol to elute the oil red O dye, and the absorbance of the oil red O at 510 nm (A510) was determined using a microplate reader.

### 4.7. Laser Confocal Microscopy

The cells were washed with PBS and then stained with Mito-Tracker green (Beyotime, Shanghai, China) at a dilution ratio of 1:3000 for 20 min. Next, the cells were treated with 4% paraformaldehyde for 5 min and were then treated with 0.2% TritonX-100 for several seconds. Subsequently, the cells were counterstained with 4′, 6-diamidino-2-phenylindole (DAPI, Solarbio, Beijing, China) for 5 min. Finally, an anti-fluorescence quenching agent (Solarbio, Beijing, China) was added dropwise and the cells were observed and photographed under a laser confocal microscope (Nikon, Tokyo, Japan).

### 4.8. Detection of the Oxygen Consumption Rate (OCR)

OCR was determined using commercially available assay kits by a Seahorse XFe24 Extracellular Flux analyzer (Agilent, Palo Alto, CA, USA). The differentiated 3T3-L1 cells were seeded on a 24-well Seahorse XF Cell Culture Microplate assay at a density of 1 × 10^4^ cells/well. Fifty percent of confluent 3T3-L1 adipocytes were treated with the EC-CM for 48 h. Then, a sensor cartridge was hydrated in Seahorse XF Calibrant at 37 °C in a non-CO_2_ incubator overnight. Prior to the assay, the treated cells were incubated with XF Base Medium (Agilent, Palo Alto, CA, USA) at 37 °C in a non-CO_2_ incubator for 1 h. OCR values were assayed under a basal condition and stress conditions exposing to oligomycin, FCCP, and rotenone/antimycin A, respectively. The results of OCR were calculated and analyzed with the Wave software.

### 4.9. Western Blotting

Tissues or cells were lysed by RIPA lysis buffer containing protease inhibitor (Pulley, Beijing, China) and the protein concentration was measured by a BCA protein assay kit (Tiangen, Beijing, China). Equal amounts of protein (30 μg) were electrophoresed on 8–12% SDS-PAGE and then transferred to PVDF membranes (Millipore, Burlington, MA, USA). The PVDF membranes were first blocked with 7% skim milk powder for 2 h and incubated overnight at 4 °C with the corresponding primary antibody solution (1:1000), followed by incubation in HRP-conjugated secondary antibody solution (1:2000) for 2 h. Finally, the signal was detected with a Gel Imaging System (Bio-Rad Laboratories, Hercules, CA, USA) with an ECL detection kit (Bayside, Shanghai, CHN). Gray-scale value analysis was performed using Image J software (NIH, Bethesda, MD, USA) to analyze the expression levels of proteins. The commercial antibodies were used in this study, including anti-UCP1 (1:1000, UCP-1), anti-PGC1-α (1:1000, both from Cell Signaling Technology, Danvers, Rockville, MA, USA), anti-PRDM16 (1:1000, Abcam, CA, USA), anti-CD31 (1:1000, Boster Biotech, Inc., Wuhan, CHN), anti-ASP (1:1000, AdipoGen), anti-p-eNOS (1:1000, p-eNOS), anti-eNOS (1:1000, eNOS), anti-p-AKT (1:1000, p-AKT), anti-AKT (1:1000, AKT), anti-p-PI3K(1:1000, p-PI3K), anti-PI3K (1:1000, PI3K), all of antibodies purchase from Cell Signaling Technology, Danvers, MA, USA and anti-β-actin (1:2000) was from FuDe Bio logical (Hangzhou, China), anti-GAPDH(1:2000, Proteintech, Inc., Wuhan, China), anti-PPARγ (1:1000, Proteintech, Inc., Wuhan, China), and anti-FASN (1:1000, Abcam, CA, USA).

### 4.10. Preparation of Conditional Medium(CM)

HUVECs and 3T3-L1 (WT and *ASP*^−/−^) were cultured in Dulbecco’s modified Earle’s medium (DMEM) containing 10% FBS, 1% penicillin/streptomycin, and heparin (100 μg/mL). When these cells grew to 95% confluence, the medium from HUVECs was collected, centrifuged at 1500 rpm and filtered with a 0.45 μm membrane to obtain EC-CM. The medium from WT or *ASP*^−/−^-3T3-L1 was collected, centrifuged at 1500 rpm and filtered with a 0.45 μm membrane to obtain WT-CM and *ASP*^−/−^-CM, respectively. These CM were stored at −20 °C until use.

### 4.11. Statistical Analysis

All data were expressed as mean ± standard error (mean ± SEM) and analyzed using GraphPad Prism 8.0 (Version number: 8.01.244). The homogeneity of variance and one-way or two-way ANOVA were performed, followed by an unpaired Student’s *t*-test, if applicable. The difference was considered significant if *p* < 0.05.

## Figures and Tables

**Figure 1 ijms-23-16166-f001:**
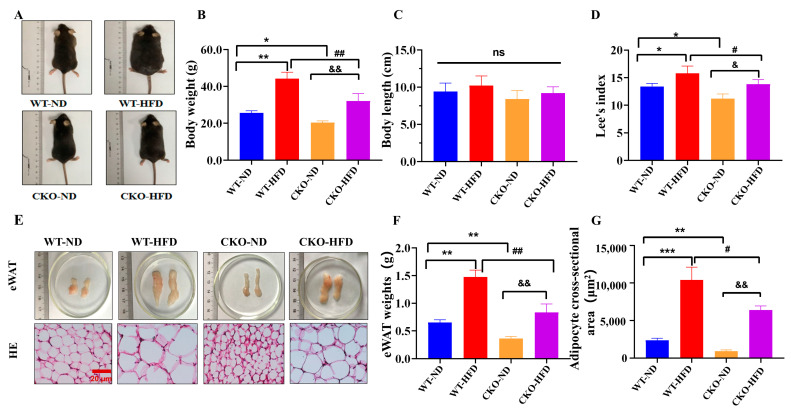
*ASP*-knockout alleviates mouse obesity induced by HFD. Representative appearance photographs (**A**) of C57BL/6 mice. Body weight (**B**), length (**C**) and Lee’s index (**D**) were measured and calculated. Representative photographs (**E**) and weights (**F**) from eWAT in mice. Representative HE staining images from eWAT (**E**) and adipocyte sizes from eWAT (**G**). Data are presented as mean ± SEM from three independent experiments (*n* = 3). Two-way ANOVA was performed followed by an unpaired two-tailed Student’s *t*-tests. * *p* < 0.05, ** *p* < 0.01, *** *p* < 0.001, vs. the WT-ND group; ^#^
*p* < 0.05, ^##^
*p* < 0.01, vs. the WT-HFD group; ^&^
*p* < 0.05, ^&&^
*p* < 0.01, vs. the CKO-ND group; ns = no significance. WT-ND: wild type mice fed with normal diet; WT-HFD: wild type mice fed with high fat diet; CKO-ND: *ASP*-conditional knockout mice fed with normal diet; CKO-HFD: *ASP*-conditional knockout mice fed with high fat diet; eWAT: epididymal white adipose tissue. Scale bars in the images of HE staining in (**E**) 20 μm. *ASP*: asprosin, HE staining: hematoxylin-eosin staining.

**Figure 2 ijms-23-16166-f002:**
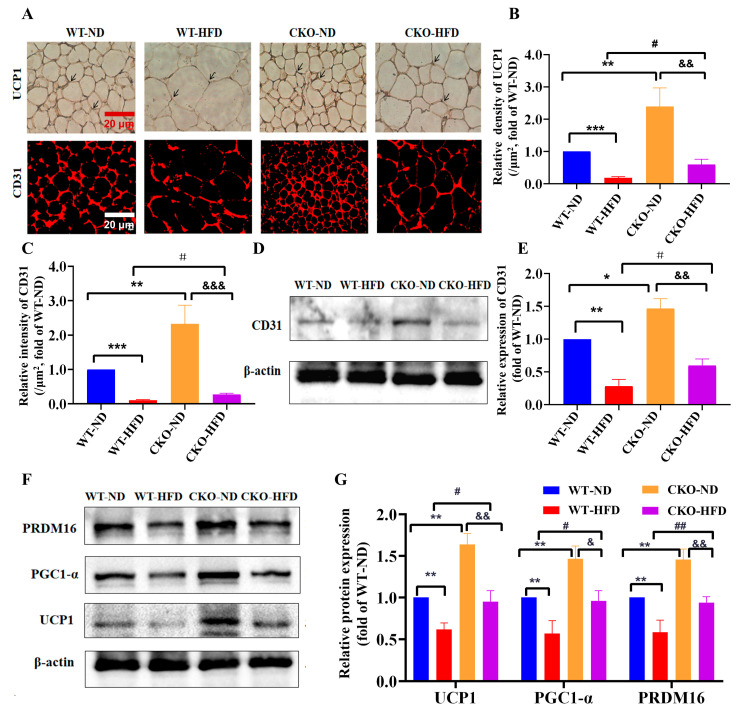
*ASP*-knockout promotes angiogenesis and browning of white adipose in vivo. Representative images (**A**) of immunohistochemistry and immunofluorescence. Relative expression levels in situ of UCP1 (**B**) and CD31 (**C**) in eWAT normalized by the adipocyte size. Expression levels of CD31 in eWAT by Western blot (**D**,**E**). Expression levels of the browning-related proteins, including PRDM16, PGC-1α and UCP-1 in eWAT (**F**,**G**). Data are presented as mean ± SEM from three independent experiments (*n* = 3). Two-way ANOVA was performed followed by an unpaired two-tailed Student’s *t*-tests. * *p* < 0.05, ** *p* < 0.01, *** *p* < 0.001, vs. the WT-ND group; ^#^
*p* < 0.05, ^##^
*p* < 0.01, vs. the WT-HFD group; ^&^
*p* < 0.05, ^&&^
*p* < 0.01, ^&&&^
*p* < 0.001, vs. the CKO-ND group. WT-ND: wild type mice fed with normal diet; WT-HFD: wild type mice fed with high fat diet; CKO-ND: *ASP*-conditional knockout mice fed with normal diet; CKO-HFD: *ASP*-conditional knockout mice fed with high fat diet; eWAT: epididymal white adipose tissue. Scale bars in the images of (**A**) 20 μm. *ASP*: asprosin, UCP1: uncoupling protein 1, PGC1-α: PPAR gamma coactivator 1, PRDM16: PRD1-BF-1-RIZ1 homologus domain-containing protein-16.

**Figure 3 ijms-23-16166-f003:**
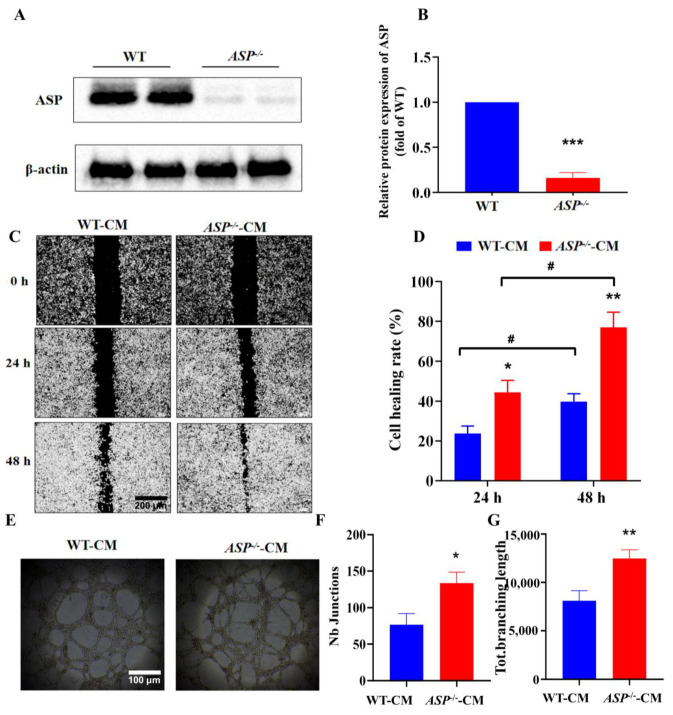
*ASP*-knockout promotes the migration and angiogenesis of vascular endothelial cells in vitro. Identification of *ASP-*knockout in 3T3-L1 adipocytes by Western blot (**A**,**B**). Wound-healing assay in exposure of HUVECs to *ASP^−/−^*-CM or WT-CM for 24 h or 48 h (**C**,**D**). Angiogenesis assay of HUVECs in vitro by a Matrigel system (**E**–**G**). Data are presented as mean ± SEM from three independent experiments (*n* = 3). Two-way ANOVA was performed, followed by an unpaired two-tailed Student’s *t*-test for Figure C and D; an unpaired two-tailed Student’s *t*-test was used in other figures. * *p* < 0.05, ** *p* < 0.01, *** *p* < 0.001, vs. the WT group or the WT-CM group; ^#^ *p* < 0.05, vs. 24 h. WT: wild type 3T3-L1 preadipocytes; *ASP^−/−^*: *ASP*-knockout of 3T3-L1 preadipocytes; WT-CM: conditional medium from WT-3T3-L1 adipocytes; *ASP^−/−^*-CM: conditional medium from *ASP^−/−^*-3T3-L1 adipocytes; Nb: numbers, Tot: total. Scale bars: 200 μm (**C**) and 100 μm (**E**). *ASP*: asprosin.

**Figure 4 ijms-23-16166-f004:**
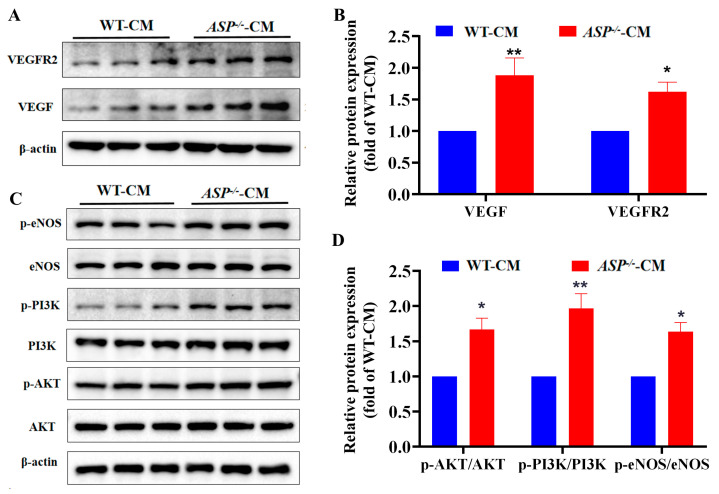
*ASP*-knockout enhances VEGF/VEGFR2 and PI3K/AKT/eNOS signaling in vitro. HUVECs were treated with *ASP^−/−^*-CM or WT-CM for 48 h and total proteins were extracted and quantified. Expression levels of VEGF/VEGFR2 pathway proteins (**A**,**B**) and PI3K/AKT/eNOS pathway proteins were analyzed by Western blots (**C**,**D**). Data are presented as mean ± SEM from three independent experiments (*n* = 3). An unpaired two-tailed Student’s *t*-test was used for the comparison between two groups. * *p* < 0.05, ** *p* < 0.01, vs. the WT-CM group. WT-CM: conditional medium from WT-3T3-L1 adipocytes; *ASP*^−/−^-CM: conditional medium from *ASP*^−/−^-3T3-L1 adipocytes. *ASP*: asprosin, VEGF: vascular endothelial growth factor, VEGFR2: vascular endothelial growth factor receptor 2; PI3K: Phosphatidylinositol 3-kinase, AKT: protein kinase B, eNOS: endothelial nitric oxide synthase.

**Figure 5 ijms-23-16166-f005:**
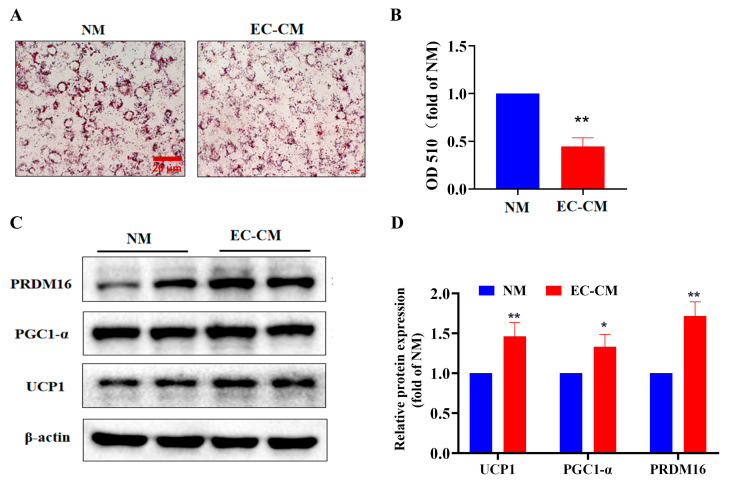
Effect of EC-CM on lipid aggregation and browning in vitro, where 80% of confluent 3T3-L1 adipocytes were treated with EC-CM or NM for 48 h. Lipid accumulation was detected by oil red O staining (**A**,**B**). Expression levels of browning-related proteins, including UCP-1, PGC1-α and PRDM16 (**C**,**D**), were analyzed by Western blots. Data are presented as mean ± SEM from three independent experiments (*n* = 3). An unpaired two-tailed Student’s *t*-test was used. * *p* < 0.05, ** *p* < 0.01, vs. the NM group. EC-CM: conditional medium from endothelial cells; NM: normal high-glucose DMEM completed medium. Scale bar in the images of (**A**): 20 μm. *ASP*: asprosin, UCP1: uncoupling protein 1, PGC1-α: PPAR gamma coactivator 1, PRDM16: PRD1-BF-1-RIZ1 homologus domain-containing protein-16.

**Figure 6 ijms-23-16166-f006:**
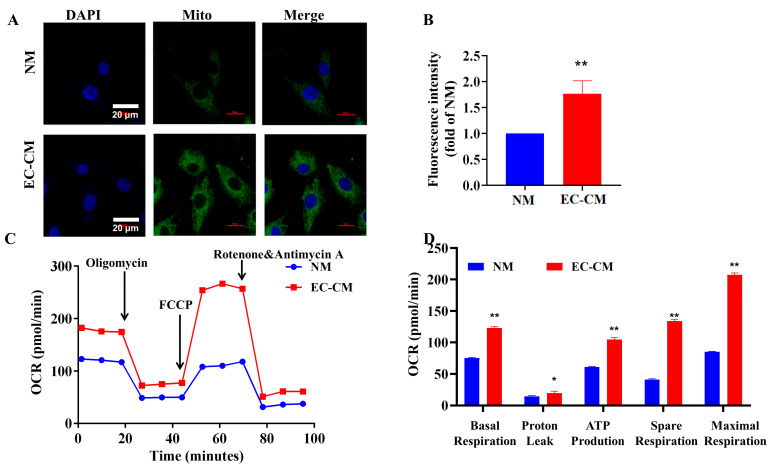
EC-CM treatment enhances the mitochondrial biogenesis and respiratory function of adipocytes in vitro. Mitochondrial contents were evaluated by a mitochondrial fluorescent probe dye under confocal microscopy (**A**,**B**). Mitochondria OCR curves (**C**) and their quantification analysis (**D**) under basal or stress conditions were prepared by a Seahorse XF Analyzer in 3T3-L1 adipocytes. Data are presented as mean ± SEM from three independent experiments (*n* = 3) and were analyzed by an unpaired two-tailed Student’s *t*-test. ** p* < 0.05, *** p* < 0.01, vs. NM group. EC-CM: conditional medium from endothelial cells; NM: normal high-glucose completed DMEM. Scale bar in the images of (**A**): 20 μm. OCR: oxygen consumption rate, FCCP: Carbonyl cyanide-p-trifluoromethoxyphenylhydrazone.

## Data Availability

All the data supporting the findings of the study are shown in this paper and are available upon reasonable request.

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
