# Peer review of "Angiogenesis–Browning Interplay Mediated by Asprosin-Knockout Contributes to Weight Loss in Mice with Obesity"

_ijms, 2022, doi:10.3390/ijms232416166_

Round 1

Reviewer 1 Report

Please refer to the attached document.

Reviewer 2 Report

General comments

It is a quiet interesting study. I believe that this manuscript tries to investigate if ASP (asprosin) could affect weight-loss in obese mice model. The results in the manuscript would help to understand the effects of ASP in vitro and in vivo model. The strength of the study study are the diversity of experimental models and a wealth of experimental results. However, there are also some of weaknesses. The following is my comments and critique.

Major

There is some over-interpretation that the effect of EC-CM between lipid accumulation and increased browning marker. The authors wanted to prove the function of asprosin, but EC-CM was tested in the cell model. I do not understand why the authors used EC-CM. Why don’t you use ASP-KO CM and WT CM from adipocyte instead of EC-CM. I think EC-CM data does not help this story.

What kind of factors in EC-CM could affect lipid level and browning marker?

Minors

1.      How to generate the adipose tissue conditioned ASP knockout mice. Please explain.

2.      What kind of HFD were used? Information (%, company, catalog no) missing.

3.      How about food intake in each group?

4.      Sometimes, ***,###,&&& not presented in figure but in legend. So, check it again and remove them in legend.

5.      Antibody information missing in Material and Method.

6.      Describe more detail method how to generate ASP knockout in 3T3-L1 by CRISPR/Cas. For example, point mutation, deletion or shifting sequence of ASP.

7.      How about the differentiation rate between WT and ASP KO in 3T3-L1?

8.      Typo in Figure 6 legend. Does not match figure and legend. 

Round 2

Reviewer 1 Report

The authors have addressed most of the previously reported concerns. Additional comments on authors' responses can be found below (Comment no.s referred to the response file submitted by the authors).

Additional comments on authors' response to comment 5 (regarding normalizing CD31 against adipocyte size):

While it is agreeable that the CD31 is an endothelial marker, calculating vascular density would involve factoring in the surface area occupied by the adipocytes. Since the adipocytes from the HFD mice are larger in size, the same area will now be occupied by fewer adipocytes, and hence this may affect the calculations of the fluorescence intensity for CD31. The authors can also normalize CD31 to the no. of adipocyte nuclei obtained from DAPI staining. It is recommended that this normalization be performed. Another way could also be to perform qPCR/WB and normalize CD31 against an adipogenic marker such as perilipin or adiponectin.

Additional comments on authors' response to comment 7 (regarding expression of VEGF in 3T3-L1 cells):

There is literature evidence suggesting the VEGF is expressed also in adipocytes in the adipose tissues. Please refer to the articles below and mentioned references:

https://www.sciencedirect.com/science/article/pii/S1550413112005013

https://academic.oup.com/endo/article/143/3/948/2989415

While the evidence of more VEGF secretion from adipocytes as a result of ASP knockout may not be necessary for this study, it could be an interesting aspect to investigate.

Reviewer 2 Report

Thanks for all your answers. All my questions are solved.

Author Response

Thanks for your positive comments.